# Modeling and Validation of Total Ionizing Dose Effect on the TSVs in RF Microsystem

**DOI:** 10.3390/mi14061180

**Published:** 2023-05-31

**Authors:** Lihong Yang, Zhumeng Li, Guangbao Shan, Qijun Lu, Yu Fu

**Affiliations:** 1School of Microelectronics, Xidian University, Xi’an 710071, China; lhyang@xidian.edu.cn (L.Y.); lizm@stu.xidian.edu.cn (Z.L.); gbshan@xidian.edu.cn (G.S.); qjlu@xidian.edu.cn (Q.L.); 2China Academy of Aerospace Standardization and Product Assurance, Beijing 100071, China

**Keywords:** total ionizing dose (TID) effect, fixed oxide charge, through-silicon via (TSV) structure, irradiation experiment, S-parameter

## Abstract

Radio frequency (RF) systems utilizing through-silicon vias (TSVs) have been widely used in the aerospace and nuclear industry, which means that studying the total ionizing dose (TID) effect on TSV structures has become necessary. To investigate the TID effect on TSV structures, a 1D TSV capacitance model was established in COMSOL Multiphysics (COMSOL), and the impact of irradiation was simulated. Then, three types of TSV components were designed, and an irradiation experiment based on them was conducted, to validate the simulation results. After irradiation, the S_21_ degraded for 0.2 dB, 0.6 dB, and 0.8 dB, at the irradiation dose of 30 krad (Si), 90 krad (Si), 150 krad (Si), respectively. The variation trend was consistent with the simulation in the high-frequency structure simulator (HFSS), and the effect of irradiation on the TSV component was nonlinear. With the increase in the irradiation dose, the S_21_ of TSV components deteriorated, while the variation of S_21_ decreased. The simulation and irradiation experiment validated a relatively accurate method for assessing the RF systems’ performance under an irradiation environment, and the TID effect on structures similar to TSVs in RF systems, such as through-silicon capacitors.

## 1. Introduction

The radio frequency (RF) circuit is an essential part of RF systems, which have been widely used in the corresponding wireless area and RF identification, including fifth-generation wireless systems, artificial intelligence, aerospace research, and so on. The RF chip plays an essential role in wireless correspondence. For example, RF transceivers minimize the radio system, and even the most conservative of the space industries identify the benefits of RF chips. With the growing markets for, and improvements in, circuit manufacturing, new technologies such as three-dimensional (3D) integration technology have been applied to RF chips. By using 3D integration technology, RF systems’ delay and power consumption can be decreased; thus, system-on-chip (SOC) with complex functions can be realized.

The effect of irradiation on RF systems cannot be ignored. Researchers are studying the impact of irradiation on complementary metal–oxide–semiconductor-integrated circuits (CMOS-ICs), metal–oxide–semiconductor field-effect transistors (MOSFETs), and the whole RF system. When studying through-silicon vias (TSVs), most researchers focus on the electrical, thermal, and stress issues of TSVs. It becomes necessary to understand the radiation resistance reliability of TSVs. In 2013, the combined effects of the total ionizing dose (TID) effect, process corners, and the temperature, on high-frequency RF circuits’ performance, were presented [1]. In 2014, back-channel implantation was introduced as an effective way to suppress the radiation-induced coupling effect [2]. In 2017, a comprehensive charge-based predictive model of interface and oxide-trapped charges in undoped symmetric long-channel double-gate MOSFETs was developed, and integrated well into all regions of operation [3]. In 2020, a new radiation-tolerant IC design method using an I-gate structure was proposed, and a device based on it was designed and fabricated in the standard CMOS process [4]. In 2021, J. Feng studied the TID effect on an 8-transistor global-shutter-exposure complementary metal–oxide–semiconductor image sensor (CIS) within a star sensor [5]. The effects of the TID Effect on CMOS-ICs are well studied [6,7,8,9,10,11,12,13]. Regarding the research on TSVs, Li Yanruoyue and his team used the finite element method to determine the effect of the thickness of the SiO_2_ layer on the distribution of thermal stress [14]. In 2021, Kan Li studied the TID effect in advanced bulk nMOS and pMOS FinFETs with nearby SiO_2_/HfO_2_ gate dielectrics and TSVs. The threshold voltage shifts introduced by TSVs are less than 25 mV [15]. Thus, the pulse frequency, oxide thickness, the radius of TSVs, and the space between TSVs have been investigated [16,17,18].

Our previous paper [19] briefly introduced the simulation in COMSOL Multiphysics (COMSOL) [20], and our analysis based on experimentation. Our current research expanded on this basis, and presented more detail. Firstly, the structure and characteristics of TSVs were introduced. Secondly, the recommended method for finite element analysis was utilized in the investigation, and TSV models were developed. Then, three types of TSV components were designed and, based on them, an irradiation experiment was designed. After the irradiation experiment, the scattering parameters (S-parameters) of TSV components were measured, while the S-parameters of three types of TSV components were also simulated in a high-frequency structure simulator (HFSS) [21]. Following all of this, the results were analyzed and discussed, and our conclusions were drawn.

## 2. Theoretical Analysis

### 2.1. Structure of TSVs

Conventional TSVs are widely used in 3D ICs. Figure 1a,b show the longitudinal section and the cross section of a conventional TSV biased in the depletion regions. The material used for the TSV core is copper, with an annular dielectric barrier typically of silicon dioxide (SiO_2_) surrounding the copper cylinder. The whole structure is built on a p-type silicon substrate.

### 2.2. TID Effect on SiO_2_

The TID effect impacts TSV in the following ways: the radiation induces excess electron–hole pairs in the insulators (Equation (1) shows the number of pairs generated per unit dose in a given oxide volume); the electrons are more mobile, and they exit, leaving hole charges; the hole charge is trapped in the oxide in deep-level traps, moving very slowly by hopping from trap to trap.
(1)$$g\left[\frac{\mathrm{ehp}}{{\mathrm{cm}}^{3}\mathrm{rad}}\right]=100\left[\frac{\mathrm{erg}}{g}\right]\left[\frac{1}{\mathrm{rad}}\right]\cdot \frac{1}{q}\left[\frac{\mathrm{eV}}{\mathrm{erg}}\right]\cdot \frac{1}{E_{p}}\left[\frac{\mathrm{ehp}}{\mathrm{eV}}\right]\cdot \mathrm{rho}\left[\frac{g}{{\mathrm{cm}}^{3}}\right]$$
where *q* = 1.6 × 10^12^ eV·erg^−1^, refers to the elementary charge; *E_p_* = 17 eV^−1^, refers to the mean ionization energy of the SiO_2_; *rho* = 2.2 g·cm^−3^, refers to the material density of SiO_2_.

Trapped holes get trapped in deep traps very close to the interface, altering the surface-carrier concentrations in the silicon. Holes and protons that move to the interface create additional defects. This trapped charge, and these interface defects, change the local carrier population, and reduce the carrier lifetime, near the surface of the SiO_2_–silicon interface.

The total density of positive fixed charge accumulated in the oxide can be calculated as a function of the total dose (*D*) using Equation (2):(2)$$\Delta N_{ot}=D\cdot g\cdot f_{ot}\cdot f_{Y}(E_{ox})\cdot t_{ox}$$

In the equation, *t_ox_* is the thickness of the oxide layer. *f_ot_* is the hole-trapping efficiency, and it can be assumed as a fitting parameter that should be empirically determined. *f_y_*(*E_ox_*) is the fractional yield. It depends on the electric field in the oxide at the moment of irradiation, and can be calculated using Equation (3):(3)$$f_{Y}\left(E_{ox}\right)=\left(\frac{\left|E\right|+E_{0}}{\left|E\right|+E_{1}}\right)\cdot m$$
where *E*_0_ = 0.1 V/cm, *m* = 0.7, and *E*_1_ = 0.55 MV/cm, for Co-60 γ-ray sources [22]. As seen in Equation (3), Δ*N_ot_* is proportional to the thickness of the oxide layer. Therefore, for TSV structures with thick oxide layers, the total density of positive fixed charge caused by radiation can be ignored.

The charge in the oxide can be calculated using Equation (4):(4)$$Q_{ot}=q\cdot \Delta N_{ot}$$

### 2.3. The Capacitance of TSVs after Irradiation

For a TSV with length *h*, *C_ox_* can be expressed by the cylindrical capacitor equation:(5)$$C_{ox}=\frac{2\pi \varepsilon_{ox}h}{\ln\left(\frac{r_{ox}}{r_{TSV}}\right)}$$
where *ε_ox_* is the permittivity of SiO_2_; *r_TSV_* is the radius of the TSV conductor; *r_ox_* is the radius of the outer surface of the TSV oxide liner.

Then, the depletion capacitance of a single TSV can be obtained by:(6)$$C_{dep}=\frac{dQ_{s}}{d\varphi \left(r_{ox}\right)}$$
where *φ*(*r_ox_*) is the surface potential at the Si–SiO_2_ interface (*r* = *r_ox_*), which can be calculated with the doping concentration of the acceptor ions, the thickness of the depletion, and *r_ox_*.

Then, the total capacitance of TSV *C_TSV_* is the series combination of the oxide and the depletion capacitance, and it can be expressed as:(7)$$C_{TSV}={\left(\frac{1}{C_{ox}}+\frac{1}{C_{dep}}\right)}^{-1}$$

The section discussed the TID effect on SiO_2_ and the capacitance of TSV after irradiation, laid the foundation for the modeling of TSV structures, and provided a simulation of the TID effect on TSV structures.

## 3. Modeling of TSV Structures and Simulation

### 3.1. Modeling of TSV Structures

The model of the conventional TSV structure was developed in COMSOL, as shown in Figure 2. The main structure consists of four essential parts: the conductive filling layer, the oxide insulating layer, the p-type silicon substrate, and the outermost shielding copper layer. The diameter of the TSV is 5 microns, and the inner insulation layer thickness is 0.5 microns.

It is assumed that charge density is similar within the lamellar at the same height in a steady state. To simplify the model, a perpendicular plane is taken in the axial direction of the TSV. Furthermore, it can be considered that the capacitance between the TSV core layer and the p-type substrate is equal everywhere along the radial direction. Therefore, the part along any radial direction can be modeled. In COMSOL, the semiconductor interface is chosen. First, a line segment and two endpoints are set to represent the structure of the inner core layer. The left endpoint is set as a thin insulating gate, whose thickness is *d_ox_*, and the surface charge density *rho_s_ox_*; and the right endpoint represents the ideal ohmic contact.

### 3.2. Simulation and Analysis

To analyze the effect of irradiation on TSV capacitance, CV curves under different radiation doses are simulated at low and high frequencies. Furthermore, to exclude the influence of oxide layer thickness, and substrate doping concentration, it is necessary to study their effects on TSV capacitance. 

#### 3.2.1. Effect of Irradiation on TSV Capacitance

The CV curves of the TSV capacitance are simulated with a p-type substrate concentration of $5\times 10^{14}{\;cm}^{-3}$, oxide thickness of 2.5 microns, frequency of 0.0001 Hz and $1\times 10^{7}$ Hz, and a total dose of 0 rad, $1.5\times 10^{4}\;rad$ and $3\times 10^{4}\;rad$, in turn.

Figure 3a shows the CV curves at low frequencies, at which the capacity of TSV is in the inverse zone. Figure 3b shows the CV curves at high frequencies, under which the inverse zone cannot form. In both Figures, after TSV is irradiated, the CV curves of the TSV capacitor drift to the left, and the higher the irradiation dose is, the more the CV curves drift to the left.

TSV is usually biased at the accumulation zone to avoid the volatility of capacitance value when used in RF chips. After radiation, the TSV capacitance enters the inverse zone at the lower voltage at both low and high frequencies.

According to Equation (5), *C_ox_* is determined by the permittivity *ε_ox_*, the length of TSV h, the radius of oxide layer *r_ox_*, and the TSV core *r_TSV_*. In the simulation and the experiment, the dimension of the TSV is constant. According to the analysis in Section 2.3, the irradiation would affect the oxide charge and permittivity of SiO_2_, leading to the increase in TSV capacitance.

#### 3.2.2. Effect of Oxide Thickness on Irradiated Capacitance

In the simulation, the substrate doping concentration is set as $5\times 10^{14}{\;cm}^{-3}$, the oxide layer is 200 nm, 1.25 micron, and 2.5 micron, the operating frequency is 0.001 Hz and $1\times 10^{7}\;Hz$, the total irradiation dose D of 0 and $3\times 10^{4}\;rad$. The simulation results are shown in Figure 4:

As shown in Figure 4a, at low frequencies, the voltage corresponding to the loss zone of the TSV capacitor is between 0 and 5 V. After the total irradiation dose of $3\times 10^{4}\;rad\;(Si)$, the CV curves of TSV shift to the left, and the thicker the oxide layer is, the more pronounced the curve deviation is. At this point, TSV capacitance enters the inverse zone at the same operating voltage. At low frequencies, the capacitance at the inverse zone is significantly greater than that in the maximum depletion zone. When designing TSV working at low frequencies and in irradiation environments, the operating point of the TSV should be set in the accumulation zone, to avoid the degradation of the TSV’s performance, caused by its instability.

Figure 4b shows the CV curves of TSV capacitance at high frequencies. Before TSV is irradiated, the maximum depletion zone of the TSV capacitor is between 0 and 5 V. At this operating voltage, the TSV capacitor demonstrates the minimum capacitance, and a relatively stable value, which is beneficial for improving the device’s performance. When the thickness of the oxide layer increases, the accumulation area capacitance of TSV decreases, and the voltage value corresponding to the maximum depletion area width increases. After the TSV capacitance is irradiated, the CV curve of the TSV capacitor moves to the left. Under standard operating voltage (0 to 5 V), the TSV capacitor is in the inverse region, and the TSV capacitance will not change significantly.

#### 3.2.3. Effect of Substrate Doping Concentration on TSV Capacitance after Irradiation

In the simulation, the thickness of the oxide layer is set as 2.5 microns, and the CV curves of TSV capacitance are simulated with substrate doping concentrations of $5\times 10^{14}{\;cm}^{-3}$ and $1\times 10^{16}{\;cm}^{-3}$, as shown in Figure 5a,b.

Figure 5a shows the CV curves of TSV capacitance at low frequencies. For TSV operating at low frequencies, the capacitance in the maximum depletion zone increases with substrate doping concentration. The maximum capacitance is obtained when TSV capacitance is in the accumulation and inverse zones, which remain the same before and after irradiation. With different substrate doping concentrations, the CV curves shift to the left by roughly the same amount after irradiation. As with Section 4.2, reducing the substrate doping concentration leads to a smaller TSV capacitance in the maximum depletion zone.

At high frequencies, the substrate doping concentration determines the capacitance in the maximum depletion zone, and is not affected by the irradiation dose.

## 4. Irradiation Experiment and Analysis

### 4.1. Design of Irradiation Experiment

In RF chips utilizing 3D integration technology, the position of the TSV affects the transmission performance of the TSV components. Three types of TSV components were designed to avoid the effect of TSV positioning. Figure 6a–c show three types of TSV components.

Figure 6 shows two main differences among those three types: firstly, the number of redundant TSVs connected to the GND; and secondly, whether the two copper layers connected to the GND are combined. The copper layers connected to the GND are separated for type A, which owns twelve redundant TSVs. The copper layers are combined for type B, which owns six redundant TSVs. The copper layers are separated for type C, which owns fourteen redundant TSVs. The dimension sets of the TSVs will be introduced in Section 5. A total of nine TSV components of each type were manufactured, to prevent the contingency of irradiation experiments on TSV components. In the irradiation experiment, three levels of irradiation dose were set, which were 30 krad (Si), 90 krad (Si), and 150 krad (Si). The irradiation source was Co-60 γ-ray. The settings of the irradiation experiment are shown in Table 1.

Three TSV components of each type were used. During the irradiation experiment, the effect of the annealing strongly affected test results. In consideration of the time between the experiment and the testing, the TSV components were introduced into the irradiation environment according to the following order. Firstly, three components were exposed to irradiation in the dose of 150 krad (Si); then, after 20 min, three components were exposed to 90 krad (Si); finally, after a further 20 min, three components were exposed to 30 krad (Si). Then, 10 min later, all TSV components were taken from the irradiation environment for testing.

### 4.2. The Results Tested before and after Irradiation

The testing environment is shown in Figure 7a,b. The equipment used for measuring the S-parameters of the TSV components were an eVue III digital imaging system (eVue III) and a vector network analyzer (VNA). During measuring, the eVue III enabled faster navigation, observation, and device measurements, which saved time. As shown in Figure 7a, two probes contact the ports of the TSV component. The probe has three ports; the middle transmits a signal and is connected to the signal TSV, and the other two ports connect to the ground. Then S-parameters are measured and shown in the VNA.

S-parameters describe the electrical behavior of linear electrical networks when undergoing various steady-state stimuli by electrical signals. Regarding a two-port system, S-parameters reflect transmission coefficients, demonstrate the transition performance of the TSV, and can be measured on the above types of equipment. For a two-port system, S_21_ and S_12_ represent the transmitting loss in the channel. The bigger S_21_ and S_12_ are, the worse the transmission performance of the TSV. S_21_ equals S_12_ because of the symmetrical structure of the TSV components shown in Figure 6.

The average S_21_ variation of the three types is shown in Figure 8a–c. The black, red, and green curves show the average S_21_ parameter variation of the three types, at the dose of 30 Krad (Si), 90 Krad (Si), and 150 Krad (Si).

### 4.3. Analysis Based on the Experiment and Measured Data

In the experiment, there were nine of each type of TSV component, making a total of 27 components. Some of the S_21_ of the same TSV components under the same irradiation dose were inconsistent in measurement. For example, at 10 GHz, the S_21_ of TSV components (type A) under the irradiation dose of 30 krad (Si) were measured after radiation. They degraded for 1.85 dB, 0.14 dB, and 0.44 dB, respectively. Under the irradiation dose of 90 krad (Si), the S_21_ of TSV components (type A) degraded for 0.86 dB, 0.06 dB, and 0.41 dB, respectively. The pressure and position of probes, process errors, and other manufacturing and testing factors caused the error. Testing a large number of TSV components and conducting statistical analysis helps to eliminate errors, whereas only 27 components have been experimented with, and tested, in the current research. The average value of the S_21_ parameter variation of the three components of the same type, at the same dose, was taken, to overcome the errors.

After taking the average, the variation trend of the S_21_ parameters became clear. The S_21_ of TSV components deteriorated after irradiation. As shown in Figure 8: with the increase in irradiation dose, the S_21_ parameters deteriorated more. At low frequencies, the average S_21_ variation at 30 Krad (Si), 90 Krad (Si), and 150 Krad (Si) were about 0.2 dB, 0.4 dB, and 0.9 dB. Although the quantities of experimental data are limited by the number of TSV components, three different types of TSV components all show the same trend. Moreover, as per the interpretation of Figure 3, because of the TID effect on TSV capacitance, the fixed charge accumulated in the oxide layer increased the capacity value at the inverse region, which caused the CV curves to drift to the left at both low and high frequencies.

According to the S_21_ simulated based on the model built in HFSS, the S_21_ curves behaved in the same manner after irradiation, which will be illustrated in the following sections.

## 5. Simulation Based on HFSS and Analysis

### 5.1. TSV Modeling in HFSS

HFSS is a full-wave 3D electromagnetic simulation software based on the finite element analysis of electromagnetic fields for microwave-engineering problems. Based on HFSS, the S-parameters of TSV components could be simulated.

According to the theoretical analysis, the total ionizing dose effect mainly affects the SiO_2_. Irradiation causes the increase in the charge in the oxide layer, and the closer that charges move to the Si–SiO_2_ interface, the greater the effect on the TSV oxide layer capacitance. According to Equation (5), the oxide layer capacitance is mainly determined by the relative permittivity of SiO_2_, the oxide layer thickness, the radius of the TSV copper core, and the length of the TSV. In the simulation, the dimensions of the TSV components were fixed, which means that the variation of TSV capacitance mainly reflected the relative permittivity of SiO_2_.

The simulation was based on type A in this section. In the TSV model of type A, the copper layers connected to the GND are separated; silicon, copper, and SiO_2_ constitute the main parts that affect the performance of the TSV component. There are twelve TSVs in the type A model, two of which are signal TSVs; the others are abundant TSVs. Each TSV has a bump beneath it, to bond the upper and lower chips. The core of the TSV is the copper column, which is surrounded by a SiO_2_ layer; the cuboid-shaped Si substrate encloses TSVs. Figure 9a,b show the flat view and the top view of the TSV model.

In the TSV model of type A, two TSVs are spaced 0.25 mm apart, the thickness of the oxide layer is 2.5 microns, the length of the TSV is 0.2 mm, the shape of the Si substrate is square, and the side length is 0.75 mm. The distance between the two signal TSVs is 0.5 mm. Modifying the size of materials will affect the simulation results, so establishing a relatively accurate model is necessary.

### 5.2. Simulation in HFSS

The relative permittivity of SiO_2_ varies from 4 to 20, and the simulation frequency range is set from 0 to 40 GHz. After the model is built and the boundary conditions are set, the S-parameters of the TSV model are simulated. The simulated S_21_ of the model of type A under different relative permittivity of SiO_2_ are as shown in Figure 10.

As shown in Figure 10, with the increasing relative permittivity of SiO_2_, the S_21_ curves move downward, and S_21_ deteriorates. The relative permittivity of SiO_2_ is set as 4, 8, 12, 16, and 20. The corresponding curves are black, red, blue, green, and yellow, and it is clear that from 5 to 25 GHz, the yellow curve is at the bottom, and the black curve is at the top.

### 5.3. Analysis Based on Simulation and Irradiation Experiment Results

As shown in Figure 10, the S_21_ of the TSV model of type A decreased rapidly with the increasing frequency within 5 GHz. In the 5–25 GHz range, the S_21_ of the TSV model of type A was nearly constant. When the frequency was greater than 25 GHz, S_21_ decreased with increasing frequency. The S_21_ curves show a similar trend under different relative permittivity of SiO_2_.

In the simulation, the relative permittivity of SiO_2_ was changed to simulate the situation after the TSV is irradiated. As shown in Figure 10, although the S_21_ curves show the same trend, the variation of S_21_ curves decreases with the exact change in relative permittivity of SiO_2_, as shown in Table 2. The data from the second to the fifth row represent the variation of S_21_ compared to the first row.

Figure 11 shows the S_21_ of the type A TSV component measured before and after the irradiation experiment, with the irradiation dose of 30 krad (Si), and the S_21_ of the type A model simulated in HFSS software, with the relative permittivity of SiO_2_ of 4 and 8.

As shown in Figure 11, within 40 GHz, the range of S_21_ of TSV components is above −2 dB, a relatively small value. According to the analysis in Section 2, the effect of increasing irradiation dose on SiO_2_ reflected the increasing relative permittivity of SiO_2_. Though irradiation would cause the deterioration of S_21_ of TSV components, the rising irradiation dose does not mean the linear change in S_21_, as shown in Figure 8. In the simulation in HFSS software, the deterioration of S_21_ caused by the shift of relative permittivity of SiO_2_ was nonlinear, and the variation trend was consistent with the previous analysis based on experimental results. The S_21_ of TSV components degraded because of irradiation, further reflecting the degradation of the performance of TSV capacitance.

## 6. Conclusions

RF systems utilizing 3D integration technology have been widely used, while the TID effect on TSVs has been neglected, hindering its application in space and in the nuclear industry. The simulation results based on a 1D model of the TSV structure showed that after irradiation, the greater the irradiation dose, the more the CV curves of the TSV capacitance drifted to the left. It was necessary to set the working point of the TSV capacitor in the accumulation zone, and to leave a certain margin for the left drift of the irradiated CV curve, to ensure the stability of the TSV capacitor. Then, three types of TSV components were designed and manufactured, and the irradiation experiment based on them was conducted. The variation trend of the S_21_ of the model in HFSS was consistent with the experimental data, and it can be concluded that the effect of irradiation on the TSV component was nonlinear. With the increasing irradiation dose, the S_21_ of the TSV components deteriorated, while the variation of S_21_ decreased. Experimenting with a large number of TSV components, and performing statistical analysis, helped to eliminate errors. In the case of insufficient data, the error was reduced by taking the average. Though the error still interfered with further quantitative analysis, the qualitative research was still very persuasive. The simulation and irradiation experiment validated a relatively accurate method for assessing the RF systems’ performance under an irradiation environment, and the TID effect on structures similar to TSV in RF systems, such as through-silicon capacitors.

## Figures and Tables

**Figure 1 micromachines-14-01180-f001:**
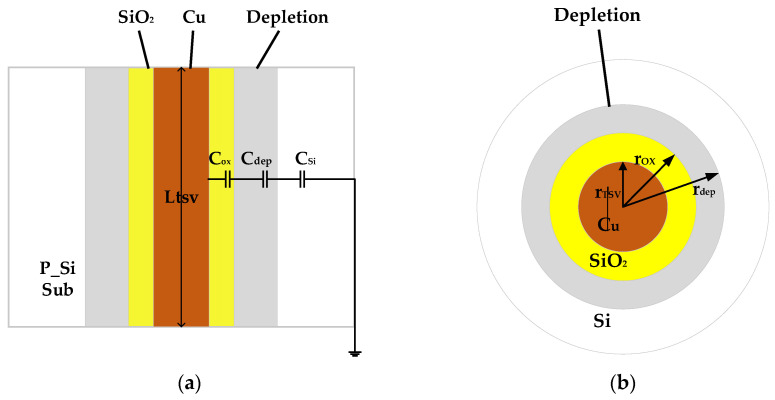
TSV architecture: (**a**) longitudinal section, (**b**) cross-section.

**Figure 2 micromachines-14-01180-f002:**
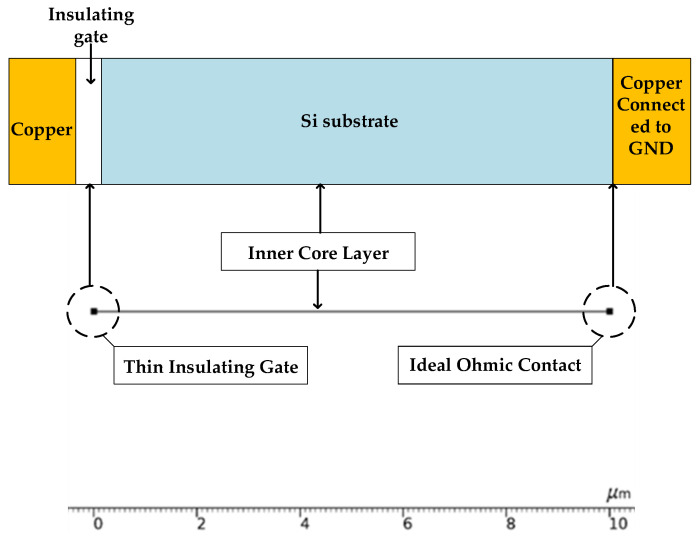
1D modeling of TSV (the p-type) in COMSOL.

**Figure 3 micromachines-14-01180-f003:**
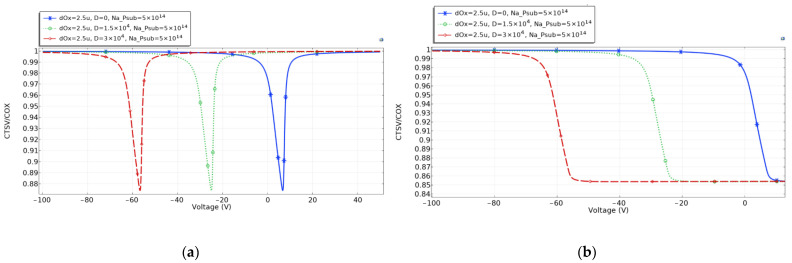
CV curve of TSV capacitance after irradiation. (**a**) low frequencies; (**b**) high frequencies.

**Figure 4 micromachines-14-01180-f004:**
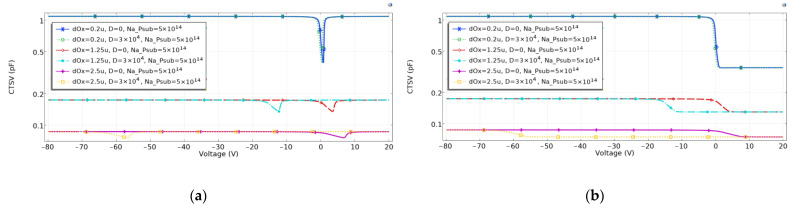
CV curves of irradiated TSV capacitances with different oxide thicknesses: (**a**) low frequencies, (**b**) high frequencies.

**Figure 5 micromachines-14-01180-f005:**
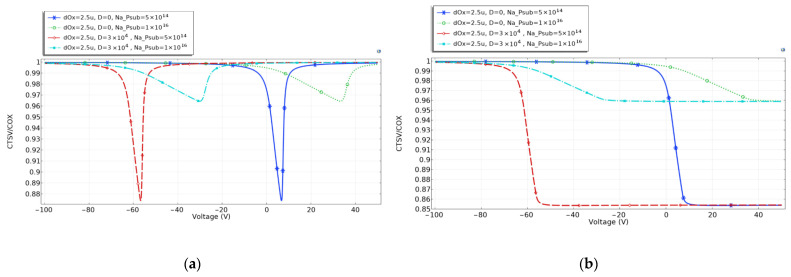
CV curve of irradiated TSV capacitances with substrate doping concentration at (**a**) low frequencies, and (**b**) high frequencies.

**Figure 6 micromachines-14-01180-f006:**
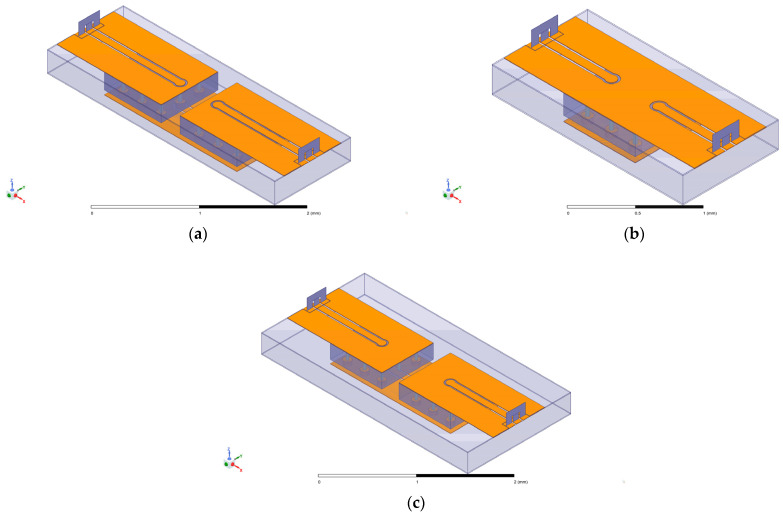
Three types of TSV components: (**a**) type A, (**b**) type B, and (**c**) type C.

**Figure 7 micromachines-14-01180-f007:**
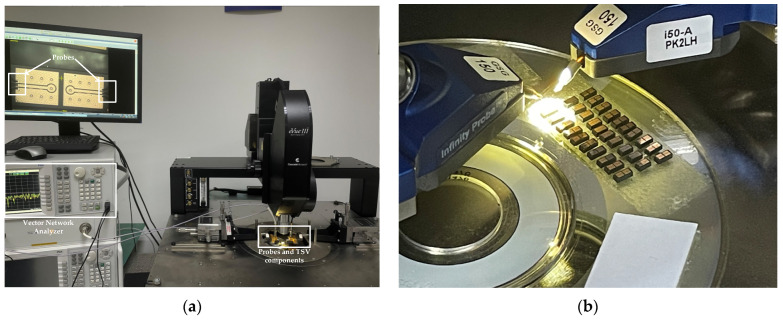
(**a**) The testing environment of the TSV components. (**b**) Probes and TSV components.

**Figure 8 micromachines-14-01180-f008:**
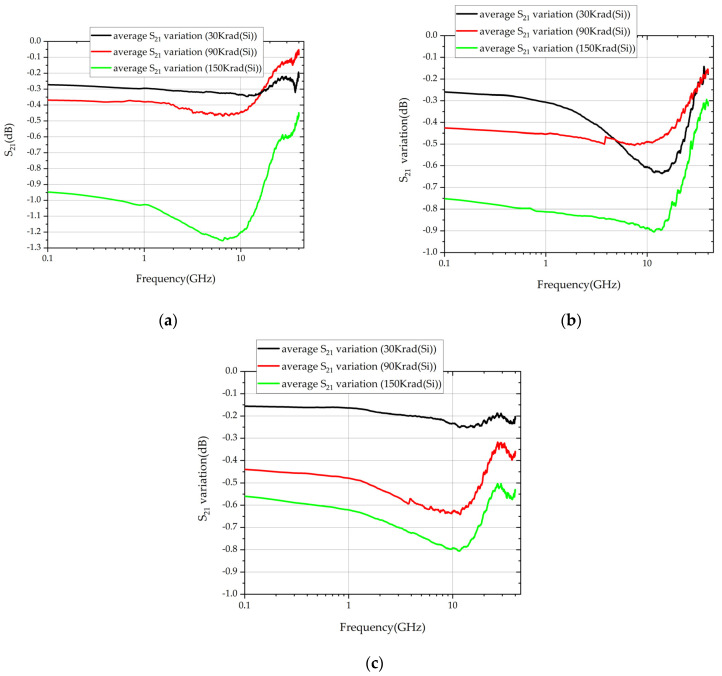
Average S_21_ variation of three types of TSV components before and after irradiation experiment: (**a**) type A, (**b**) type B, (**c**) type C.

**Figure 9 micromachines-14-01180-f009:**
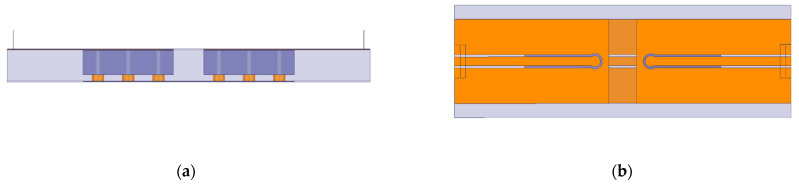
TSV model of type A modeled in HFSS: (**a**) flat view of the TSV model; (**b**) top view of the TSV model.

**Figure 10 micromachines-14-01180-f010:**
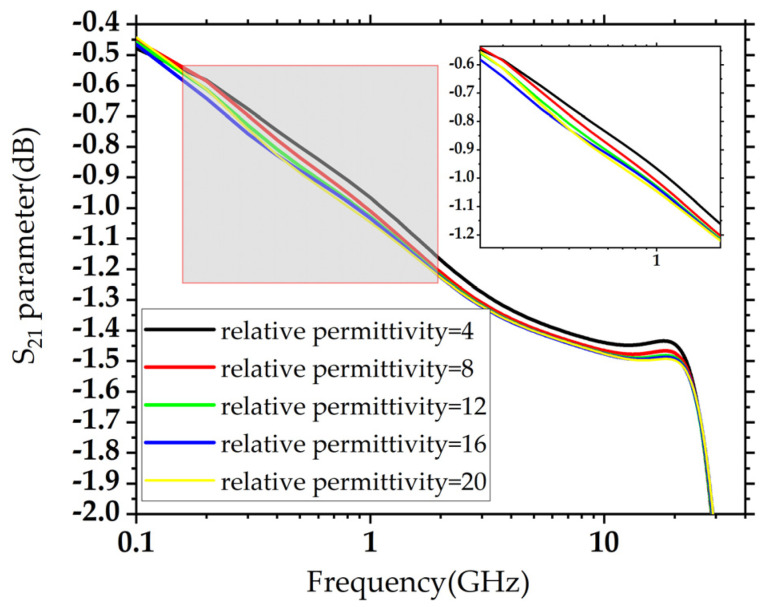
S_21_ simulated in HFSS, based on the TSV model of type A.

**Figure 11 micromachines-14-01180-f011:**
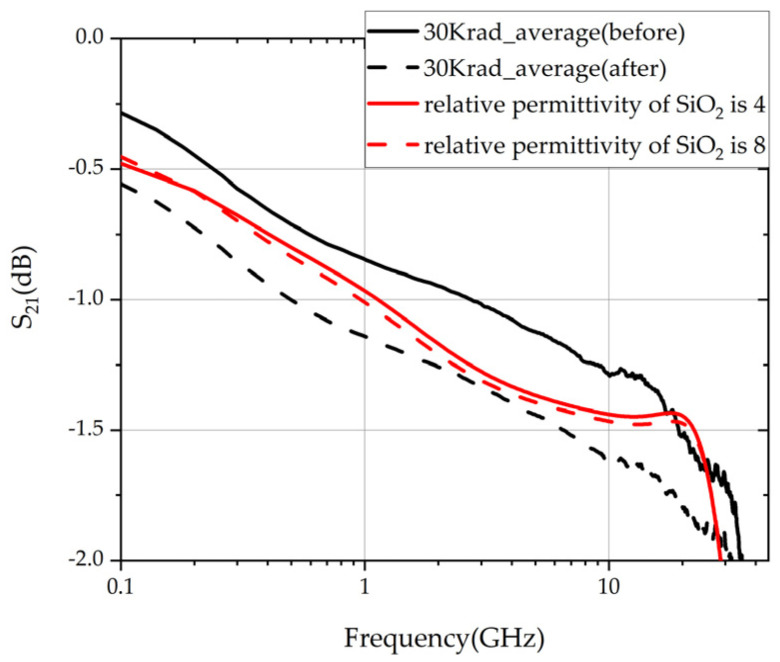
S_21_ of TSV component of type A.

**Table 1 micromachines-14-01180-t001:** Experimental conditions of different irradiation doses.

Irradiation Dose	Experimental Conditions	
	Irradiation Rate (rad/s)	Irradiation Time (min)
30 krad (Si)	50	10
90 krad (Si)	50	30
150 krad (Si)	50	50

**Table 2 micromachines-14-01180-t002:** Variation of S_21_ at different frequencies, and at different relative permittivities of SiO_2_.

The Relative Permittivity of SiO_2_	1 GHz	5 GHz	10 GHz	15 GHz	20 GHz
4	−0.96656	−1.36706	−1.43972	−1.44308	−1.44511
8	−0.04377	−0.02657	−0.02608	−0.03188	−0.02888
12	−0.06195	−0.03464	−0.03526	−0.046	−0.04344
16	−0.06711	−0.03547	−0.03693	−0.05096	−0.04852
20	−0.08022	−0.03451	−0.03634	−0.05446	−0.05253

## Data Availability

The data presented in this study are available on request from the corresponding author. The data are not publicly available due to the regulations of the research institutions.

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
