# Peer review of "Modeling and Validation of Total Ionizing Dose Effect on the TSVs in RF Microsystem"

_micromachines, 2023, doi:10.3390/mi14061180_

Round 1

Reviewer 1 Report (Previous Reviewer 1)

Accept in present form.

Author Response

Reviewer 2 Report (Previous Reviewer 3)

The captions and axes on your figures are not readable.

You need to give some explanation why you are simulating CV curves up to 100V.  Is that relevant?  These CV curves from your model are easily calculated by hand in 1-D, no simulations are necessary.  The results will depend on your Nit and Not assumptions. 

It is difficult to learn anything meaningful from your s-parameter measurements.  Is the variability due to measurement problems?  Device variations?  Is it signficant at all or in the noise?  This needs more analysis.

English structure isn't too bad.  My main comment is that some of the phrases are a little "awkward".  By that, I mean that after reading it a couple of times, I know what is intended to be said, but it isn't always obvious the first pass through.

Author Response

Reviewer 3 Report (New Reviewer)

Authors have presented work on " Modelling and Validation of total ionizing dose effect on TSVs in RF Microsystems. I have few questions or suggestions prior to accept it. 

1. Effect of neighboring TSV's  not analyzed. I suggest take pitch between TSV's as per ITRS road map then study the total ionizing dose effect as well

2. While studying effect of irradiation on TSV  capacitance using COMSOL, what are different physics incorporated?

3. If liner material changed to high K dielectric material how would it effect ?

4. Future scope of the study need to be incorporated in the conclusion section.

5. Add some recent literatures. 

Author Response

Reviewer 4 Report (New Reviewer)

The paper is about the evaluation of the TSVs under the effect of total ionizing dose. The paper has merit and it is interesting. But there are a number of issues that should be resolved. I have the following comments (MA -> major, MI -> minor).

1) MA. The texts in the labels, in the axis, and in the legends of Figs.3, 4 and 5 are too small in size. Axis labels and legends are unreadable. Indeed, the reader cannot understand what the curves refer to. Plots must be improved to make them readable.

2) MA. Figure 10 contains a subplot without any label in the axis. I can understand that the subplot may be a magnification of the region evidenced in Fig. 10, but the subplot must at least contain the values on the axis.

3) MI. Please define explicitly the abbreviations at the first occurrence, e.g. HFSS.

4) MI. In the abstract, the S21 parameter is not described or defined.

5) MA. In general, the references may be extended, in particular in the Introduction section. As you also introduced the TID effects in CMOS transistors, some good references could potentially be:

[1] F. Faccio, et al., “Radiation-induced short channel (RISCE) and narrow channel (RINCE) effects in 65 and 133 nm MOSFETs,” IEEE Trans. Nucl. Sci., vol. 62, no. 6, pp. 32663-3270, Dec. 2015.

[2] C.M. Zhang, et al., “Characterization of GigaRad Total Ionizing Dose and Annealing Effects on 28-nm Bulk MOSFETs,” IEEE Trans. Nucl. Sci., vol. 64, no. 10, pp. 2639-2647, 2017.

[3] T. Ma, et al., “TID degradation mechanisms in 16 nm bulk FinFETs irradiated to ultra-high doses,” IEEE Trans. Nucl. Sci., vol. 68, no. 8, pp. 1571-1578, Aug. 2021.

[4] S. Bonaldo, et al., “TID effects in highly scaled gate-all-around Si nanowire CMOS transistors irradiated to ultra-high doses,” IEEE Trans. Nucl. Sci., vol. 69, no. 7, pp. 1444-1452, July 2022.

[5] M. Gailardin, et al.. “Radiation Effects in Advanced Multiple Gate and Silicon-on-Insulator Transistors”. IEEE Trans. Nucl. Sci., vol. 60, no. 3, pp. 1970-1991, June 2013.

[6] D. M. Fleetwood, “Radiation Effects in a Post-Moore World,” IEEE Trans. Nucl. Sci., vol. 68, no. 5, pp. 509-545, May 2021.

6) MA. I suggest inserting more details about the experimental tests and the experimental setup used to measure the TSV structures. What type or radiation was used? What is testing setup?

7) MA. The y-axis scale of Figs. 8 and 11 does not seem to be appropriate, as the curves appears to overlap one to each other. I suggest to magnify the y-axis in order to zoom in the differences between the curves.

8) MI. In order to evidence the difference in the capacitance values between the different curves of Fig. 4, I suggest to change the y-axis scale to logarithmic.

9) MA. Line 339. The text reports "Though experimental error makes it harder to proceed with further quantitative analysis, the conclusion is definite: irradiation affects TSV capacitance strongly." Looking to the experimental results it does not seem that the radiation induced large shifts in the capacitance values of TSVs. Could you please explain where this large difference is visible?

10) Line 248. Could you add a brief explanation of what is the S parameter?

The text contains many typos. Please carefully check the text. There are many missing spaces, extra no-required full stops, and missing capital letters.

For example, line 10, missing space in "frequency(RF)"; lines 149 and 151 report "Figure.3" with extra full stop; caption of Fig. 8 reports many missing spaces after the figure letters; caption of Fig. 7 contains missing spaces and missing capital letters. Many others are along the manuscript.

English could be improved, as many sentences are a repetition of previous ones. In general, I suggest applying more synthesis to express the results and discussion. There are sections that are too long because the writing is very dispersive, e.g. Introduction section.

Round 2

Reviewer 2 Report (Previous Reviewer 3)

My main issue that remains with the paper is summarized in Section 4.3.  The information is not presented clearly, but it really sounds like you are simply omitting data that doesn't fit with your hypothesis.  Best I can understand it, the 30k type-A parts range from 0.14 to 1.85 dB changes in loss.  These simply cannot be thrown out.  Instead, we need to know how repeatable these measurements are in the first place.  This is what a control part is for.  These are being probed.  Probes get dirty.  Aligment can be off.  If you make the same measurement multiple times, how much variation will you see?  At the end of the paper, you're trying to convince us that changes smaller than the experimental differences you are seeing are signficant.  However, what you are showing doesn't support that. 

I suggest presenting all of your data.  Perhaps a table of the data at various frequencies.  Highlight the outliers.  Let the readers see what the differences are.  Convince them that your measurements are correct and repeatable.  You can't make statements like "Though experimental error makes it harder to proceed with further quantitative analysis, the conclusion is definite: the effect of irradiation on TSV capacitance is not negligible." without being able to present a strong argument based in the data that this is indeed correct.

Also, it is impossible to distinguish between the different pre- and post- curves in figure 8.    Figures 10 and 11 aren't much better, since you are using similar colors on the same plot.

You probably need references for COMSOL and HFSS

Better

Author Response

Reviewer 4 Report (New Reviewer)

Thank you for following my comments. The paper has gained quality. I do not have further technical comment.

There  is still a number of typos in the text. Example:

- Line 140, "Figure.3 (a)" -> "Figure (a)"

- Labeling of Figure 8, (c) is indicated as copyright symbol instead of (c)

Round 3

Reviewer 2 Report (Previous Reviewer 3)

N/A

None

Author Response

This manuscript is a resubmission of an earlier submission. The following is a list of the peer review reports and author responses from that submission.

Round 1

Reviewer 1 Report

This paper presents an original work on TID effects on 3D integration technology for RF applications . Analysis and underlying assumptions are well described and properly verified. Finally, this paper is well organized and well written and therefore is suggested for publication with minor revisions.

I think some of the figures are not efficient to show the validity and efficiency of the proposed approach. Some of them need to be replotted with a higher quality.

In addition, reviewer thinks that the bibliography should be extended and references should be made. Authors may cite other relevant papers and books to provide a more comprehensive background of  TID effects on FETs and double-gate FETs for interested readers in the introduction:

F Jazaeri, CM Zhang, A Pezzotta, C Enz, "Charge-based modeling of radiation damage in symmetric double-gate MOSFETs", IEEE Journal of the Electron Devices Society 6, 85-94

Li, Z.; Elash, C.J.; Jin, C.; Chen, L.; Xing, J.; Yang, Z.; Shi, S. Comparison of Total Ionizing Dose Effects in 22-nm and 28-nm FD SOI Technologies. Electronics 2022, 11, 1757. https://doi.org/10.3390/electronics11111757

S. Jagannathan et al., "Sensitivity of High-Frequency RF Circuits to Total Ionizing Dose Degradation," in IEEE Transactions on Nuclear Science, vol. 60, no. 6, pp. 4498-4504, Dec. 2013, doi: 10.1109/TNS.2013.2283457.

...

An introduction, and conclusion as a summary of the paper should be short and precise to easily capture the interest of a potential reader.

Reviewer 2 Report

Dear authors, thank you for giving me the opportunity to read your paper.

The article presents a simulation study on the effect of total ionizing dose (TID) irradiation on Through Silicon Vias (TSV) structures; it also compares these simulations to some experimental data.

Some general comments.

First, I suggest the authors to improve the English language to help the reader in the reading of the paper.

While simulations can be accepted, I think that any comparison between simulations and experimental data is not really possible. Experimental data are affected by a huge uncertainty (Figure 8). The claim in the abstract "In this paper, we present a simulation methodology [...] which could save a lot of time and costs on irradiation experiments" is too strong and not supported by results.

Some more specific remarks:

In the Introduction, references cited are not complete, since they refer either to radiation damage on different kind of structures (not TSV), or to different kind of damages (thermal stress). Any reference on other studies of the effect of TID on TSV, which is the topic of the paper, is missing. 

In Section 2, equation 4 is a replication of equation 3, while the proper expression for the charge in the oxide is missing.

Figure 2 is basically empty 

Figure 3 mixes the effect of the different doses and of the different substrate doping concentration (not addressed in 4.1 but addressed later in 4.3)

The testing procedures are not described properly and it is not clear to me  what Figure 7 is showing.

 Experimental data are affected by a huge uncertainty (Figure 8) and are presented only for one of the three models used. I think that any kind of comparison (qualitative and quantitative) between simulations and data is not possible or should be better justified. I would at least extend the comparison to the other 2 models (B and C). 

For these reasons (lack of clearness, uncomplete results), I would not suggest the publication of the paper in the present form

Reviewer 3 Report

Units for radiation dose are krad(SiO2) or krad(Si), not Krad.  A lower-case "k" is the SI prefix for kilo.

Paragraphs 2 and 3 in the introduction, and all associated references, look to be irrelevant.  They seem to have nothing to do with TID in TSVs.

Page 3, line 106 - incorrect symbol used for rho.

Page 3 ,line 119 - I don't think this is the correct reference. 

Why is equation 3 repeated as equation 4?

Fig 3 legend and scales are unreadable.

Why are simulation doses different from measured doses?

How were traps modeled in simulations?  What energies?  Where were they located?

Round 2

Reviewer 2 Report

Dear Authors

Thank you for the revised version you have prepared. 

Unfortunately I feel that the changes you have implemented are only formal and nothing has been added on the scientific side. In addition, I strongly suggest you to have your paper read by a native English speaker before a further submission for review.

This would take time, I know, but I don't see any other way to improve the overall quality of the paper.

Thank you again and regards

Reviewer 3 Report

English still needs revision.

References in introduction are not relevant.  They just seem to be included for the sake of having something to include.

References for equations unless you have derived them, in which case they should be included in an appendix.  

Line 105 - you haven't shown this at all.  Where is is Nit shown?

why do a 1D in comsol?  Why do cylindrical analytically?  Why not do 1D analytically to have a direct comparison?

I don't know what I'm supposed to see in Fig 2.

Equations in the model DC.  Where is the frequency / time component?  I don't see anything in your model that would cause the capacitance to vary with frequency.

Axes and legends are unreadable.  too small.

Table 1 has the default caption.

What irradiation source?  What measurement equipment?

Errors and inconsistencies in the data need to be addressed or shown.  How do you know they are wrong?  

I don't understand the purpose of Table 2.